# Training Effects on the Stress Predictors for Young Lusitano Horses Used in Dressage

**DOI:** 10.3390/ani12233436

**Published:** 2022-12-06

**Authors:** Clarisse S. Coelho, Ana Sofia B. A. Silva, Catarina M. R. Santos, Ana Margarida R. Santos, Carolina M. B. L. Vintem, Anderson G. Leite, Joana M. C. Fonseca, José M. C. S. Prazeres, Vinicius R. C. Souza, Renata F. Siqueira, Helio C. Manso Filho, Joana S. A. Simões

**Affiliations:** 1Equine Academic Division, Faculty of Veterinary Medicine, Lusofona University (ULHT), Campo Grande 376, 1749-024 Lisbon, Portugal; 2Mediterranean Institute for Agriculture, Environment and Development, Universidade de Évora, 7006-554 Évora, Portugal; 3Center for Studies, Extension and Research in Equidae, Universidade Federal da Bahia (NEEPEq-UFBA), Salvador 40170-110, Bahia, Brazil; 4Faculdade de Medicina Veterinária, Universidade Federal de Santa Maria (UFSM), Santa Maria 97105-900, Rio Grande do Sul, Brazil; 5Núcleo de Pesquisa Equina, Universidade Federal Rural de Pernambuco (UFRPE), Recife 51171-900, Pernambuco, Brazil; 6CIISA-Centre for Interdisciplinary Research in Animal Health, Faculty of Veterinary Medicine, University of Lisbon, 1300-477 Lisbon, Portugal; 7Associate Laboratory for Animal and Veterinary Science (AL4AnimalS), Faculty of Veterinary Medicine, University of Lisbon, 1300-477 Lisbon, Portugal

**Keywords:** cortisol, equine, equestrian performance, field tests

## Abstract

**Simple Summary:**

Dressage is an Olympic equestrian discipline that requires specific characteristics such as agility and obedience skills and a precise interaction between the horse and the rider. The comprehension of such a level of effort and planning a specific training protocol are important and tracking the exercise-related stress responses of the horses represents a way to monitor sports training and the wellbeing of athletic animals. The purpose of this study was to investigate stressful responses during a 6-week training protocol in young Lusitano dressage horses. Nine 4-year-old horses were evaluated before and after six weeks of a training protocol which included 40–80 min of individually intensity-adjusted preparatory exercises for dressage, six times per week. For both moments, the horses were examined before and immediately after the dressage simulation tests, and at 30 and 240 min during the recovery period. The evaluated stressors included heart rate (HR), heart rate variability (HRV), cortisol, total white blood cell count (WBC), and neutrophil and lymphocyte counts. After training, there were significant reductions in cortisol, HR, total WBC, neutrophils, and lymphocytes, and an increase in the HRV parameters related to a cardiac vagal modulation. In conclusion, the chosen training protocol led to better fitness. Such data can be used to evaluate performance, but also to predict the welfare of athletic horses.

**Abstract:**

The purpose of this study was to investigate stressful responses during a 6-week training protocol in young Lusitano horses used for dressage. The hypothesis was that the proposed training protocol would improve fitness and ensure the welfare of the animals by reducing stress predictors. Nine 4-year-old horses were evaluated before (M1) and six weeks after (M2) beginning a training protocol. The training program was performed six times per week and included 40–80 min of individually intensity-adjusted preparatory exercises for dressage. For both moments, the horses were examined before (T0) and after (T1) dressage simulation tests (DST), and at 30 (T2) and 240 min (T3) during the recovery period. Blood samples were taken to determine the horses’ cortisol levels, total WBC, and neutrophil and lymphocyte counts. All variables were analyzed by one-way ANOVA and Tukey tests, with *p* ≤ 0.05. After training, there was a significant reduction in cortisol (*p* = 0.0133), HR (*p* = 0.0283), total WBC (*p* < 0.0001), and neutrophil (*p* < 0.0001) and lymphocyte (*p* = 0.0341) counts. Other findings included an increase in HRV parameters related to a cardiac vagal modulation. In conclusion, the chosen training protocol led to better fitness as the horses worked more intensively with lower cardiovascular requirements, and they showed blunted cortisol responses at M2. Such data can be used to evaluate performance, but also to predict the welfare of athletic horses.

## 1. Introduction

Stress promotes necessary modifications to physiological mechanisms to obtain homeostasis after an event of physical origin, such as physical exercise, fatigue, and lesions, or after an event of psychological origin, such as fear and anxiety [1,2,3,4]. Afterward, the sequence of events involves a behavioral response, sympathetic–adrenal–medulla response, and hypothalamic–pituitary–adrenal cortex response [3], with the release of many hormones, with an emphasis on cortisol and adrenaline [5,6]. The main purpose of exercising horses is to enhance the energy mobilization of the nervous system and the muscles during activity [2,5,7,8]. Furthermore, cardiovascular effects are observed, with increases in heart rate (HR), stroke and systolic volume [4,5], and an effort-dependent splenic contraction that enhances tissue perfusion through a transient increase in circulating red blood cells (RBC) [9,10].

Besides RBC elevation, stressful events such as physical activity can lead to leukocytosis and changes in other biomarkers related to inflammation [11]. The mobilization of white blood cells (WBC) occurs in different ways. While cortisol induces neutrophilia and lymphopenia (and an increase in the N:L relation), known as stress leukogram, adrenaline promotes the mobilization of peripheral neutrophils and lymphocytes to the circulating pool, known as physiological leukogram [2,10,12]. According to Snow et al. [13], cortisol increased the WBC count of 10–30% of thoroughbred horses during exercise.

Competition, high intensity training, and traveling induce stress, both acute and chronic [3,6,14]. Stress is not always harmful, and positive stress (“eustress”) indicates body accommodation and adaptation [14]. Thus, tracking the exercise-related stress response of horses represents a way to monitor the sports training [15,16] and wellbeing of athletic animals [14]. The typical stress determination parameters include cortisol, HR, and heart rate variability (HRV) [3]. HRV is an indicator of the response of the autonomic nervous system to stress [8], and it can indicate a stress response independent of physical activity [17].

Generally, a training protocol is individually planned and considers the aims of the athlete and the sport itself, aiming to achieve maximum physical conditioning [18,19] for the practice of a specific sport/event. It is characterized by the systematic and continuous practice of physical exercises, with a gradual increase in intensity, intercalated by rest periods [20,21]. In humans, physical activity benefits both emotional and cognitive wellbeing [21]. Thus, recently, emotional factors must also be considered during preparation for exercises or competitions in order to ensure the welfare of the horses and people involved [4].

Currently, the study of good training practices represents a gap in equine sports medicine, especially in equestrian disciplines such as dressage [17] that require quite peculiar and specific characteristics such as agility, balance, obedience, skills, and a precise interaction between the horse and rider [20,22]. This interaction is so intense that previous studies have indicated a certain synchrony in the HR [23] and cortisol release [14] of both the horse and its rider.

The training of Lusitano horses typically begins when they are 3 years old, and this initial learning period can be significantly challenging for the horses. Therefore, the purpose of this present study was to quantify the stress predictor responses (cortisol, HR, HRV, and leukogram) to a 6-week training protocol in young Lusitano horses used in the equestrian discipline of dressage. The hypothesis of the authors was that the proposed training schedule would improve the athletic fitness of the studied Lusitanos horses, which would be reflected in a reduction in the stress predictors.

## 2. Materials and Methods

This project was approved by the Ethics and Animal Welfare Committee of Lusofona University (CEBEA—ULHT, number 112/2021).

Animals: Nine Lusitano horses were enrolled in the present study, with an average weight of 473.9 ± 44.0 kg and an average age of 4 years old. The animals were considered healthy following a previous clinical evaluation and regular hematology and biochemical analyses at rest. The animals belonged to an equine training center in Lisboa (38°49′12.2″ N, 9°13′16.5″ W) and they were kept under identical husbandry conditions. Their diets consisted of perennial ryegrass (*Lolium perenne*) hay, which they were fed twice each day (according to the weight, corporal score, and level of physical effort of each individual), along with a commercial concentrate (13% crude protein per 1.0 kg/100 kg body weight) divided into three daily portions (9 a.m., 1 p.m. and 6 p.m.). Water was always available. All horses had been exercising regularly and training for dressage practice for at least 6 months before being enrolled. Briefly, the dressage training began at the end of each horse’s third year of life and consisted of going through an initial training phase (tame) which aimed to familiarize the horse with the new experiences of this new phase of life. This phase was marked by leaving the field and the process of getting used to living in stables (boxes), as well as accepting and interacting with horseback riding work. Initially, flat groundwork (hygiene [brushing and bathing] and rider–horse interactions) and lunging were introduced, followed by a posteriori simple saddle riding work, and, finally, riding work, properly speaking (execution of the first basic exercises of low school [circles and changes of hand]). In their weekly routines, the horses were worked five to six times, with an individual training program specific to each animal, which was modified according to individual needs. These programs covered management actions, snaffle bit work, ride work, and “in-hand” work.

Physical training program: All horses were exposed to a 6-week interval training schedule that consisted of 40 to 80 min of exercises, six days per week. In this period, they performed preparatory activities for dressage events (specific effort), with moderate intensity (resistance effort) alternated with moments of high intensity exercise (strength effort). The activities always respected the horses’ individual limits (personality and physical responses) and met the individualized learning needs of the proposed activity for each animal. For the rest of the day, preferably at the opposite time of day of the work, the animals were released into an outdoor paddock for 50 min. On Sundays, no physical activities were scheduled. This training schedule was performed by the same horse rider, and it is summarized in Table 1.

Dressage Simulation Tests (DSTs): The horses were submitted to two DSTs, one before (M1) and another after (M2) completing the 6-week training protocol described in Table 1. Each rider and horse performed a sequence of typical physical exercises of dressage modality, which mainly included transitions; circles; extended walks, trots, or canters; stride length variations; diagonals; and straight lines and serpentines. The DSTs were performed in a 30 × 15 m indoor riding arena, demarcated by letters in accordance with FEI (Fedération Equestre Internationale) and APSL (Portuguese Association of Lusitano Horses Breeders) regulations. Before being ridden, each horse was warmed up with lunging exercises for 20 min, followed by 15 min of saddle ride work to properly prepare for the DSTs and reach maximal relaxation. All the physical trials were performed in a 15 × 30 m covered arena with a dry sand ground, in the morning (between 7 a.m. and 11 a.m.), during the months of August and September (summer season in the Northern Hemisphere). The local mean temperature was 19.7 °C and the mean relative humidity ~83.8%. All horses were ridden by the same experienced rider, and according to the welfare precepts of Coelho et al. [24]. The mean weight of the riders was 62.1 kg, and for the leather saddles, it was 8.2 kg, and the accessories weighed an estimated 0.9–1.2 kg.

Heart rate, HRV and speed monitoring: For the entire duration of the DSTs, the horses used an integrated heart rate (HR) and GPS monitoring system (M430, Polar Electro, Lake Success, NY, USA) which recorded RR intervals every 1 s. Later, the data were transferred and analyzed using the Polar Flow software (Polar Electro, Lake Success, NY, USA). The horses’ HRVs were analyzed with Kubios HRV software (Biomedical Signal Analysis Group, Kuopio, Finland). From the RR intervals, the following time-domain measures were obtained: mean RR interval, standard deviation (SD) of the RR intervals, root mean square (rMSSD) of the successive differences in the RR intervals, NN50 (number of successive RR interval pairs that differed by more than 50 ms), pNN50 (relative number of successive RR interval pairs that differed by more than 50 ms) and HRV triangular index (the integral of the RR interval histogram divided by the height of the histogram). The frequency-domain analysis was performed using the Fast Fourier Transform method with the sampling frequency set at 8 Hz. The power in the heart rate spectrum was divided into three different frequency bands: very low-frequency power (VLF, 0 to 0.04 Hz), low-frequency power (LF, 0.04 to 0.15 Hz) and high-frequency power (HF, 0.15 to 0.4 Hz). The non-linear properties of the horses’ HRVs were studied through a Poincaré plot (SD1 describing the short-term variability, SD2 describing the long-term variability, and SD2/SD1 ratio) and approximate entropy (ApEn), which provided a measure of the irregularity of the signal. Detrended fluctuation analysis (DFA) was performed to determine the long-range correlations in the non-stationary physiological time series, yielding both short-term fluctuation (α1) and long-term fluctuation (α2) slopes.

Stress biomarkers evaluation: Heart rates and blood samples were taken from the jugular vein of each horse at T0 (at rest, before exercise), T1 (immediately after the test), T2 (at 30 min during recovery), and T3 (at 240 min during recovery). For leukogram analysis, samples were stored in test tubes containing EDTA (Vacutainer, BD), and they were processed with an automated hematology analyzer (IDEXX ProCyte Dx, IDEXX Laboratories, Haarlemmermeer, The Netherlands) within 4–6 h. Cortisol evaluation was determined in samples obtained at T0, T1, and T3, and test tubes containing clot activators were centrifuged (Hettich Universal 320 R, Hettich Labs, Bäch, Switzerland). Refrigerated serum was transported to the laboratory using a cooler with ice. Samples were kept in a −80 °C freezer (PHCbi Ultra-Low Temperature Freezer MDF-DU502VH) until determination through a chemiluminescence assay (IMMULITE^®^ 2000 Immunoassay System, Siemens, Erlangen, Germany).

Statistical analysis: The results were analyzed using the PROC MIXED program (SAS 9.1, SAS Institute Inc., Cary, NC, USA). All data were evaluated for normality using the Kolmogorov–Smirnov test. Analyses of variance for repeated measures, followed by comparisons between means (*t*-test and Tukey test), were carried out to evaluate the possible influences of the exercise tests (DSTs) at M1 and M2 and the possible influences of the training schedule on all the stress indexes (M1 vs. M2). The results were expressed as means and standard deviations, and values of *p* < 0.05 were considered significant. The Pearson correlation test was used to evaluate the strengths of association between the total WBC, neutrophil and lymphocyte counts, HR, and cortisol; strong correlations occurred when the r^2^ vale ranged from 0.7–0.99 and *p* < 0.05.

## 3. Results

The DSTs significantly influenced cortisol (Table 2), HR (Table 3) at M1 and M2 and total WBC and neutrophil and lymphocyte counts only at M1 (Table 4).

Significant differences were observed between M1 and M2 for all parameters, with significant reductions after training for the cortisol (Table 2), HR (Table 3), and leukocyte parameters (Table 4). The HRV indexes are described in Table 5, in which it is possible to observe a significative improvement in all components related to vagal activity.

Strong positive correlations (*p* < 0.05) were observed between total WBC and neutrophil count (r^2^ = 0.78) and total WBC and lymphocyte count (r^2^ = 0.70). However, a moderate positive correlation between HR and cortisol (*p* < 0.05; r^2^ = 0.57) was also observed. 

According to the riders, no signs of discomfort or reduced performance were observed during exercise execution throughout the six weeks of training and DSTs. The tests lasted approximately 5 min (±32.6 s), and each animal covered a mean distance of 220 m.

## 4. Discussion

The hypothesis proposed by the authors regarding the reduction in cortisol levels and other stress predictors due to an imposed training protocol was confirmed. Previous studies on humans and animals have shown that the more trained an individual is, the more attenuated the cortisol release is, along with greater energy efficiency and a consequent mental and physiological adaptation to the level of imposed physical effort [7,8,21,25,26].

Regular exercises are recommended to induce significant metabolic stress, with physiological modifications that aim to enable the work capacity and sports performance of horses for a specific sport modality [5,10]. Moreover, stress responses decrease with repeated exposure to the same challenge [15,21]. In the present research, Lusitano horses showed reductions in cortisol values at M2, signaling a possible reduced sensitivity of the adrenal gland as a result of the successful adaptation to the training program [18,21], along with the positive stress (“eustress”) resulting from the exercise practice [14,20,27]. In classical equitation horses, a reduction in cortisol after training has been described [7,15], making a direct relationship between improved fitness (or stress) and lower cortisol levels. After an initial stress, an adaptation to the new situation occurs, accompanied by a reduction in cortisol release [15], as was observed for these Lusitano horses.

Contrarily, other authors have described increases in cortisol levels in response to training [14,20]. However, in the latter study’s experimental protocols, the animals were evaluated during competition, while in the present research, the Lusitano horses were evaluated during a routine day of physical activity, which involved far fewer stressors. Despite this difference, Cayado et al. [20] observed a major discrepancy between the basal cortisol values and those recorded after competition in less experienced animals when studying training effects on show jumping and dressage horses. In addition, Ferraz et al. [19] demonstrated an increase in cortisol when training Arabian horses on a treadmill, attributing this finding to a possible non-adaptation to the imposed physical effort or intensification of exercise (fartlek training).

Besides lower values at M2, the DSTs led to a minor significant increase in serum cortisol (~58.3%) when compared to those observed at M1 (~35.5%). Nonetheless, these values were considerably minor compared to those observed for show jumping and dressage horses (190%) [14] and endurance horses (206%) [10]. Again, these animals were evaluated during a competition with many stressors to be considered, such as transport, novelty, a noisy public, music, and rider stress. Cortisol is crucial for exercise practice as it promotes substrate mobilization, which enhances the activity of glucagon and catecholamines (with increases in liver gluconeogenesis, liver ketogenesis, and lipolysis and decreases in lipogenesis), and behavior modifications [17] that help animals deal with the challenge of exercise/competition [5,7]. Therefore, this transient cortisol release is expected and must happen, as deficient levels of cortisol will affect performance [3,7,15]. Cortisol release occurred after dressage practice, regardless of the presence of spectators [8], as well as for other equine disciplines, such as marcha (4-beat gait) [28] and jumping [6]. As cortisol release is dependent upon the intensity and duration of the exertion, the lower increase at M2 after the same intensity DSTs reinforced the beneficial modifications after completing 6 weeks of training, with a greater experience for the horses [3] since the lower cortisol concentrations were associated with increased learning efficiency [4].

During these 6 weeks, the intensity of the exercises and the duration of each sequence were adjusted according to the response of each individual animal. Respecting the horses’ individual limits was crucial for physical improvement that did not generate discomfort or stress that could lead to a new increase in cortisol or even physical injuries and overtraining. It is known that when a load increases too much, either in duration or intensity, it generates an expected increase in cortisol to mobilize more energy and compensate for the new stress due to the physical activity [3,19,21], which may compromise a horse’s welfare [29]. Other causes of the increase in serum cortisol may have been fasting or a poor diet prior to exercise [7]. However, the increase in intensity was not reflected in the serum cortisol, likely due to the animals’ adaptation to the progressively imposed load. The interaction of the rider–horse relationship here was proven to be crucial for the former to perceive the horse’s reactions and better balance the load without compromising good performance on the next day of activities [8].

It is worth mentioning the importance of rest, which is fundamental for muscle recovery accompanied by anabolic effects [30]. In addition, moments of relaxation were scheduled in which the animals were released in a paddock in the period opposite to that established for training.

Along with cortisol evaluation, HR and HRV (beat-to-beat) are physiological parameters used to evaluate stress [17]. Training promotes physiological changes in the cardiorespiratory system so that the body can respond positively to the level of imposed exercise. Mainly, these changes are observed as reductions in HR, increases in heart volume, and higher systolic volumes with greater efficiency in cardiac contractility, which contributes to less cardiac effort when performing a physical effort of the same intensity [31]. Therefore, the HR reductions observed for the Lusitano horses occurred due to reduced physical and psychological stress [8,17,32] because the animals adapted to the routine of the activities, as well as to the alterations in the expression of B-adrenergic receptors in cardiac muscle [33].

The DSTs increased HR, although at a lesser magnitude at M2, as observed for cortisol, reflecting the changes in the autonomic activity on sinusal intrinsic modulations [31]. This rise typically occurs to meet the greater physical and energy demand of active working muscles and the need for the dissipation of metabolic by-products and heat [5,6,34]. Nonetheless, the recovery of the horses’ HRs to pre-exercise levels (in the stable) within 30 min denotes the good athletic conditioning of all the horses used in the present experimental protocol [9,34], which was established even before the setting of the training schedule.

While increases in HR are primarily caused by physical effort, decreases in HRV indicate a stress response [8]. A low rMSSD has been shown to be a reliable marker of fatigue and poor recovery in endurance horses after exercise [35]. During the experimental period, increases in the values of the HRV variables SDRR and rMSSD reflected a shift towards parasympathetic dominance [15]. Most studies on human athletes have suggested that high HRVs are associated with increased fitness levels [36,37]. Further, Ille et al. [6] described higher rMSSD values in experienced jumping horses. The training protocol, therefore, caused a more pronounced cardiac vagal modulation in the horses, leading to better fitness. Results in horses are controversial. Nyerges-Bohak et al. [16] described an increase induced by training in standard Hungarian racehorses, as observed for human elite athletes [38], in a so-called parasympathetic saturation, a physiological phenomenon not yet studied in horses.

There are few studies that directly address the use of information obtained from a leukogram in equine sports medicine. As measured at M1, the DSTs led to leukocytosis (17.4%), determined by neutrophilia (9.8%) and lymphocytosis (25%), accompanied by a reduction in the N:L ratio. Such findings are compatible with the release of leukocytes from the marginal compartment into the circulating pool under the influence of adrenaline, a typical reaction to acute stress [2]. A greater increase (45%) in the number of total leukocytes was described for Quarter horses used in vaquejada [11] and for endurance horses [10], and it was attributed to cortisol and adrenaline. However, no significant changes were generated by the DSTs at M2 for the same hematological variables, in addition to being lower in relation to those at M1. This is consistent with supposedly lower stress, whether possibly physical or even psychological, corroborating the responses observed for cortisol and HR.

Given that the cortisol and WBC values decreased during the training period, we can verify an adaptation to the stimulus [10,21], and thus, the group had adapted to the physical effort imposed, reducing the resulting physical, and possibly psychological, stress.

## 5. Conclusions

The results showed reductions in all stress predictors studied (cortisol, HR, HRV, and WTC count), indicating that the imposed 6-week training protocol was efficient in inducing adaptations and promoting fitness, while also guaranteeing the horses’ welfare (eustress). These data must be also considered when evaluating the wellbeing of dressage horses.

## Figures and Tables

**Table 1 animals-12-03436-t001:** Training schedule for the young Lusitano horses.

Week	Training Goals
1	Groundwork: Warm-up—lunging (15 min) Saddle work exercises (20–30 min): Relaxation exercises Familiarization with the indoor ridding arena Cardiovascular exercises—with and without rider (5 min, 1–2x/week)
2	Groundwork: Warm-up—lunging (15 min) Saddle work exercises (20–30 min): Relaxation exercises Gait exercises (rectitude and regularity training) Exercises focusing on circles and curves Cardiovascular exercises—with and without rider (5 min, 1–2x/week)
3	Groundwork: Warm-up—lunging (15 min) Saddle work exercises (20–30 min): Relaxation exercises Gait exercises (rectitude, regularity, and cadence training) Cardiovascular exercises—with and without rider (5 min, 2–3x/week)
4	Groundwork: Warm-up—lunging (15 min) Saddle work exercises (20–30 min): Relaxation exercises Exercises focusing on circles and curves Gait exercises (regularity training) Cardiovascular exercises—with and without rider (5 min, 2–3x/week)
5	Groundwork: Warm-up—lunging (15 min) Saddle work exercises (20–30 min): Relaxation exercises Gait exercises (energy and amplitude training at a walk, trot, and canter) Cardiovascular exercises—with and without rider (5 min, 2–3x/week)
6	Groundwork: Warm-up—lunging (15 min) Saddle work exercises (20–30 min): Relaxation exercises Exercises focusing on circles and curves Gait exercises (changes in amplitude and energy) Stretching and collection exercises Cardiovascular exercises—with and without rider (5 min, 2–3x/week)

**Table 2 animals-12-03436-t002:** Cortisol (nmol/L) measured in the dressage simulation tests used to evaluate a 6-week training protocol for Lusitano horses.

Cortisol, nmol/L	Experimental Period			*p*	
	T0	T1	T3	DST	M1 × M2
M1	72.04 ± 14.09 ^b^	114.04 ± 31.88 ^a^	40.88 ± 21.65 ^c^	<0.0010	0.0133
M2	56.79 ± 37.85 ^b^	76.99 ± 26.63 ^a^	47.51 ± 22.80 ^b^	0.0096	

Note: different letters in the same line denote significant differences by Tukey test (*p* < 0.05). T0: at rest; T1: immediately after the exercise; T3: at 240 min during recovery.

**Table 3 animals-12-03436-t003:** Heart rate measured in the dressage simulation tests used to evaluate a 6-week training protocol for Lusitano horses.

Parameters	Experimental Period				*p*	
	T0	T1	T2	T3	DST	M1 × M2
HR, bpm						
M1	38.8 ± 17.2 ^b^	76.4 ± 12.4 ^a^	41.1 ± 4.8 ^b^	39.8 ± 4.4 ^b^	<0.0001	0.0283
M2	28.9 ± 4.5 ^b^	71.0 ± 14.8 ^a^	34.4 ± 2.9 ^b^	38.1 ± 11.8 ^b^	<0.0001	

Note: different letters in the same line denote significant differences by Tukey test (*p* < 0.05). T0: at rest; T1: immediately after the exercise; T2: at 30 min during recovery; T3: at 240 min during recovery.

**Table 4 animals-12-03436-t004:** Total WBC and neutrophil and lymphocyte counts measured in the dressage simulation tests used to evaluate a 6-week training protocol for Lusitano horses.

Parameters	Experimental Period				*p*	
	T0	T1	T2	T3	DST	M1 × M2
WBC, ×10^3^/μL						
M1	8.87 ± 0.76 ^b^	10.41 ± 0.69 ^a^	9.23 ± 0.90 ^b^	9.69 ± 0.92 ^ab^	0.0028	<0.0001
M2	7.59 ± 1.57	8.35 ± 0.88	7.97 ± 1.29	8.72 ± 1.64	0.4077	
Neutrophils, ×10^3^/μL						
M1	4.18 ± 0.30 ^b^	4.59 ± 0.43 ^a^	4.33 ± 0.48 ^ab^	4.75 ± 0.30 ^ab^	0.0182	<0.0001
M2	3.41 ± 0.85	3.51 ± 0.56	3.36 ± 0.47	4.12 ± 1.58	0.3840	
Lymphocytes, ×10^3^/μL						
M1	4.32 ± 0.66 ^b^	5.40 ± 0.76 ^a^	4.50 ± 0.66 ^ab^	4.38 ± 0.85 ^b^	0.0130	0.0341
M2	3.91 ± 0.96	4.54 ± 0.63	4.32 ± 1.10	4.13 ± 0.80	0.5424	

Note: different letters in the same line denote significant differences by Tukey test (*p* < 0.05). T0: at rest; T1: immediately after the exercise; T2: at 30 min during recovery; T3: at 240 min during recovery; WBC: total white blood cell count; Neutrophils: neutrophils count; Lymphocytes: lymphocytes count.

**Table 5 animals-12-03436-t005:** Heart rate variability measured in the dressage simulation tests used to evaluate a 6-week training protocol for Lusitano horses.

Time-Domain Results			
Variable	M1	M2	*p*
Mean RR (ms)	509.9 ± 1.54 ^b^	623.6 ± 1.20 ^a^	0.0010
SDNN (ms)	185.5 ± 1.18 ^b^	295.8 ± 1.32 ^a^	0.0000
rMSSD (ms)	233.7 ± 1.51 ^b^	378.4 ± 1.43 ^a^	0.0000
NN50 (beats)	84.0 ± 0.68 ^b^	112.54 ± 0.63 ^a^	0.0030
pNN50 (%)	13.89 ± 1.10 ^b^	23.77 ± 1.24 ^a^	0.0000
RR triangular index	10.88 ± 2.70 ^b^	16.98 ± 1.15 ^a^	0.0030
**Frequency-Domain Results**			
	**M1**	**M2**	** *p* **
LF (%)	56.78 ± 1.17	53.26 ± 1.13	0.6600
HF (%)	36.83 ± 1.20	39.13 ± 1.20	0.6260
LF/HF Ratio	2.51 ± 0.20	3.39 ± 0.5	0.6730
**Nonlinear Results**			
	**M1**	**M2**	** *p* **
Poincaré plot			
SD1	165.40 ± 1.07 ^b^	270.20 ± 1.30 ^a^	0.0000
SD2	202.80 ± 1.30 ^b^	291.30 ± 1.26 ^a^	0.0460
SD2/SD1	1.37 ± 0.55	1.71 ± 0.60	0.5790
Approximate entropy (ApEn)	0.45 ± 0.15	0.58 ± 0.19	0.1470
DFA			
α1	0.71 ± 0.28	0.82 ± 0.19	0.5530
α2	0.75 ± 0.18	0.66 ± 0.18	0.3950

Note: different letters in the same line denote significant differences by *t*-test (*p* < 0.05). M1: before training; M2: after 6 weeks of training.

## Data Availability

The data presented in this study are available on request from the corresponding author.

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
