# Peer review of "Training Effects on the Stress Predictors for Young Lusitano Horses Used in Dressage"

_animals, 2022, doi:10.3390/ani12233436_

Round 1

Reviewer 1 Report

Training effects on stress predictors of young Lusitano horses 2 used in dressage

 The paper aimed to investigate the stressful  responses of young Lusitano horses used for dressage.  The topic is original, since Competition, high-intensity training, and traveling induces stress, both acute and chronic.

Statistical analysis

It is better to ad statistical models with details.

Add the correlation between all measured parameters.

The result  and discussion sections highlight the relationship between physiological parameters

The conclusions must be improved

Add tables of correlations

Author Response

Dear Reviewer and Editor.

Thanks of all considerations. Here are our answers. 

Sincerely,

The Authors.

Comments and Suggestions for Authors – REVIEWER 1

Training effects on stress predictors of young Lusitano horses used in dressage

 The paper aimed to investigate the stressful  responses of young Lusitano horses used for dressage.  The topic is original, since Competition, high-intensity training, and traveling induces stress, both acute and chronic.

Thanks for your considerations.

Statistical analysis

It is better to ad statistical models with details. Modifications were made.

Add the correlation between all measured parameters. Correlation tests were done.

The result  and discussion sections highlight the relationship between physiological parameters. Okay.

The conclusions must be improved. Modifications were made.

Add tables of correlations. We included it on the text.

Reviewer 2 Report

The aim of the study was to determine the training effects on stress predictors of young Lusitano horses used in dressage.

The approach of the study appears very original. The contents of the manuscript are quite interesting by his methodology and through the tools of quantification used. The manuscript reads smoothly and is easy to understand.  The aims, scope, and results of the study are clearly stated.  I have very much enjoyed reading this paper. I find it interesting and clearly written, and satisfying also all the other publication criteria of the journal Animals. The study provides a very valuable addition to this line of research, and adds relevantly to the subject with additional original findings. I thus find that this paper definitively delivers results that will surely be of interest to the readership of the journal Animals.  I recommend the publication of this paper after another English reading and minor corrections.

However, the introduction and discussion should be improved. It is mostly connection with lack of information included in most recent publication. Most of cited publications (80%) are older than 20 years, thus more recent should be discussed because more wide knowledge is contained in the recent literature .

Corrections:

L29- total leucocytes – it should be total white blood count (WBC) because in other parts Authors are using WBC

Introduction

L55 – The authors should add information about Eustress. The physical exercise, if well organized, determines forms of adaptation that improve performance and may be called as a “Eustress”. “Eustress” implies that “correct or optimal stress level” may have a positive impact on welfare. Stress response after race and endurance training sessions in horses was documented to be rather as eustress. Such information should be added and citation as a proof for that.

L67 – not only basic hematological parameters are shown to be changed during training. Also activity of shite blood cells such increased proliferation, differentiation, activity, and reactive oxygen species production in racehorses, thus such information should be added and citation. Also, anti-inflammatory state connected with changes in anti-inflammatory cytokines is created during training which was documented in endurance as well as race horses during training as well as changes in acute phase proteins such as serum amyloid A (SAA). SAA is considered to be a biomarker in exercise horses. Thus also this information should be added with related citation.

L66 - Stress leucogram is produced by cortisol influence, whereas adrenalin induce excitation which influence on leucogram as well. Adrenalin influence on spleen contraction and changes in leucogram may be also influenced by that.

L74 – also other non-invasive techniques are considered to be used as monitoring methods in sport horses. Recently, infrared thermography may be used because it correlates with lactate concentration in blood during race training in horses.

Materials and methods

L167 – the information that samples were always obtained at the same time should be added because changes in cortisol concentration are circadian rhythm depended.

L168 - Also the storage temp. should be added.

L175 – also the producent of chemiluminescence assay should be added

Results

Table 2 – ad units also to the table

Discussion

L231 – the information that during conditioning the cortisol contentration after the training is decreasing. As well as the horses with better fitness level have got lower increase in cortisol concentration after exercise.  

L234 – the eustress should be discussed because optimal stress in essential for training progress.

L285 – during conditioning also β2-adrenergic receptors on the peripheral blood mononuclear cells influencing on proliferation, phenotype, functions, and reactive oxygen species production are changing in horses. Thus, training influences on several mechanisms in sport horse organism. Also probably this receptors in cardiac muscle may be changed during conditioning. Such information should be included.

References

Most of cited publications (80%) are older than 20 years, thus more recent should be added.

Author Response

Dear Reviewer and Editor.

Thanks of all considerations. Here are our answers. 

Sincerely,

The Authors.

Comments and Suggestions for Authors – REVIEWER 2

The aim of the study was to determine the training effects on stress predictors of young Lusitano horses used in dressage.

The approach of the study appears very original. The contents of the manuscript are quite interesting by his methodology and through the tools of quantification used. The manuscript reads smoothly and is easy to understand.  The aims, scope, and results of the study are clearly stated.  I have very much enjoyed reading this paper. I find it interesting and clearly written, and satisfying also all the other publication criteria of the journal Animals. The study provides a very valuable addition to this line of research, and adds relevantly to the subject with additional original findings. I thus find that this paper definitively delivers results that will surely be of interest to the readership of the journal Animals.  I recommend the publication of this paper after another English reading and minor corrections.

Thanks for your considerations!!

However, the introduction and discussion should be improved. It is mostly connection with lack of information included in most recent publication. Most of cited publications (80%) are older than 20 years, thus more recent should be discussed because more wide knowledge is contained in the recent literature .

Corrections:

L29- total leucocytes – it should be total white blood count (WBC) because in other parts Authors are using WBC. Modifications were made.

Introduction

L55 – The authors should add information about Eustress. The physical exercise, if well organized, determines forms of adaptation that improve performance and may be called as a “Eustress”. “Eustress” implies that “correct or optimal stress level” may have a positive impact on welfare. Stress response after race and endurance training sessions in horses was documented to be rather as eustress. Such information should be added and citation as a proof for that. Modifications were made (lines 71-72).

L67 – not only basic hematological parameters are shown to be changed during training. Also activity of shite blood cells such increased proliferation, differentiation, activity, and reactive oxygen species production in racehorses, thus such information should be added and citation. Also, anti-inflammatory state connected with changes in anti-inflammatory cytokines is created during training which was documented in endurance as well as race horses during training as well as changes in acute phase proteins such as serum amyloid A (SAA). SAA is considered to be a biomarker in exercise horses. Thus also this information should be added with related citation. Modifications were made keeping the focus on the leukogram and stress markers.

L66 - Stress leucogram is produced by cortisol influence, whereas adrenalin induce excitation which influence on leucogram as well. Adrenalin influence on spleen contraction and changes in leucogram may be also influenced by that. Thanks for the observation. That was our point to differentiate cortisol and adrenaline actions.

L74 – also other non-invasive techniques are considered to be used as monitoring methods in sport horses. Recently, infrared thermography may be used because it correlates with lactate concentration in blood during race training in horses. Thanks again. We just did not mention that to focus on stress markers.

Materials and methods

L167 – the information that samples were always obtained at the same time should be added because changes in cortisol concentration are circadian rhythm depended. This was described in line 145.

L168 - Also the storage temp. should be added. This was described in line 177.

L175 – also the producent of chemiluminescence assay should be added. Information was added.

Results

Table 2 – ad units also to the table. Modifications were made.

Discussion

L231 – the information that during conditioning the cortisol contentration after the training is decreasing. As well as the horses with better fitness level have got lower increase in cortisol concentration after exercise. It was discussed in lines 224-227 and 233-234.

L234 – the eustress should be discussed because optimal stress in essential for training progress. Modifications were made.

L285 – during conditioning also β2-adrenergic receptors on the peripheral blood mononuclear cells influencing on proliferation, phenotype, functions, and reactive oxygen species production are changing in horses. Thus, training influences on several mechanisms in sport horse organism. Also probably this receptors in cardiac muscle may be changed during conditioning. Such information should be included. Modifications were made trying to keep focus again on stress markers.

References

Most of cited publications (80%) are older than 20 years, thus more recent should be added. We included recent papers. From the 36 citations, 20 are from the last 10 years (55.5%), 10 are from 2002-2012 (27.8%) and only 6 of 36 from before 2002. These are considered reference papers (Hyyppa, 2005).

Round 2

Reviewer 1 Report

I recomend the publication of the paper

Author Response

Thanks for your considerations!
